# Genome-Wide Circular RNA Expression Patterns Reflect Resistance to Immunomodulatory Drugs in Multiple Myeloma Cells

**DOI:** 10.3390/cancers13030365

**Published:** 2021-01-20

**Authors:** Theresa Jakobsen, Mette Dahl, Konstantinos Dimopoulos, Kirsten Grønbæk, Jørgen Kjems, Lasse Sommer Kristensen

**Affiliations:** 1Department of Biomedicine, Aarhus University, Høegh-Guldbergs Gade 10, DK-8000 Aarhus, Denmark; theresa@biomed.au.dk; 2Department of Hematology, Rigshospitalet, Copenhagen University Hospital, Blegdamsvej 9, DK-2100 Copenhagen, Denmark; mette.dahl.01@regionh.dk (M.D.); konstantinos.dimopoulos@regionh.dk (K.D.); kirsten.groenbaek@regionh.dk (K.G.); 3Biotech Research and Innovation Centre, BRIC, Copenhagen University, Ole Maaløes Vej 5, DK-2200 Copenhagen, Denmark; 4Department of Molecular Biology and Genetics (MBG), Aarhus University, C.F. Møllers Allé 3, DK-8000 Aarhus, Denmark; jk@mbg.au.dk; 5Interdisciplinary Nanoscience Center (iNANO), Aarhus University, Gustav Wieds Vej 14, DK-8000 Aarhus, Denmark

**Keywords:** immunomodulatory drugs, multiple myeloma, circular RNA, epigenetics, genome-wide profiling, RNA-sequencing, non-coding RNA

## Abstract

**Simple Summary:**

Multiple myeloma (MM) constitutes the second most common hematological malignancy and is caused by aberrant plasma cell proliferation in the bone marrow. While recent improvements in the treatment of MM has been observed using immunomodulatory drugs (IMiDs), patients often relapse due to acquired drug resistance and no cure for the disease is currently available. In this report, we profile circular RNA (circRNA) expression patterns in cultured MM cells being sensitive to IMiDs and their resistant counterparts. CircRNAs constitute a large class of non-coding RNA molecules with emerging roles in cancer development and progression, but have not previously been explored in this context. We found that global circRNA expression patterns reflect IMiD sensitivity, but the most downregulated circRNA in IMiD resistant MM cells did not seem to be a direct driver of IMiD resistance. Future studies should investigate other circRNA candidates identified here in the context of IMiD resistance.

**Abstract:**

Immunomodulatory drugs (IMiDs), such as lenalidomide and pomalidomide, may induce significant remissions in multiple myeloma (MM) patients, but relapses are frequently observed and the underlying molecular mechanisms for this are not completely understood. Circular RNAs (circRNAs) constitute an emerging class of non-coding RNAs with important roles in cancer. Here, we profiled genome-wide expression patterns of circRNAs in IMiD-sensitive MM cells and their resistant counterparts as well as in IMiD-resistant cells treated with specific epigenetic drugs alone or in combination. We found that genome-wide circRNA expression patterns reflect IMiD sensitivity and ciRS-7 (also known as CDR1as) was the most downregulated circRNA upon acquired resistance. The depletion of ciRS-7 correlated with increased methylation levels of the promoter CpG island of its host gene, LINC00632. Expression of LINC00632 and ciRS-7 was partly restored by treatment with a combination of an EZH2 inhibitor (EPZ-6438) and a DNA methyl transferase inhibitor (5-azacytidine), which also restores the IMiD sensitivity of the cells. However, knockdown of ciRS-7 did not affect IMiD sensitivity and we found that ciRS-7 also becomes epigenetically silenced after prolonged cell culture without drug-exposure. In conclusion, we found that genome-wide circRNA expression patterns reflect IMiD sensitivity in an in vitro model of acquired resistance.

## 1. Introduction

Multiple myeloma (MM) is a malignancy of terminally differentiated monoclonal plasma cells that accumulate in the bone marrow. It is the second most common hematological malignancy and accounts for nearly 2% of all cancer cases in the United States [1], and the incidence of MM is globally increasing due to an ageing population [2]. Newly introduced treatments, including proteasome inhibitors, epigenetic drugs (epi-drugs), monoclonal antibodies and immunomodulatory drugs (IMiDs), have significantly prolonged the survival expectancy of MM patients [3]. Although patients usually respond to initial therapy, most will eventually succumb to the disease [4], and no clinically validated biomarkers are currently available to predict response to IMiD therapy [5,6].

The IMiDs (thalidomide, lenalidomide and pomalidomide) have been shown to function, in part, through binding to the protein cereblon (CRBN), which is part of an E3 ubiquitin ligase complex. When bound by IMiDs, the complex changes its affinity for many of its targets, including the transcription factors, Ikaros (encoded by *IKZF1*) and Aiolos (encoded by *IKZF3*), which upon ubiquitination are degraded by the proteasome [7,8,9]. Ikaros and Aiolos contribute to cancer development by enhancing the expression of IRF4 and c-MYC [10]. Ikaros degradation also results in increased production of IL-2 within T cells, leading to increased numbers of functional cytotoxic T cells, which may cause an immunogenic assault on the cancer cells [7,9]. However, while the degradation of Ikaros and Aiolos occurs within hours after lenalidomide treatment, it can take several days for the MM cells to die, indicating that CRBN may have additional targets that play a role in the IMiD-induced cell death [11]. Protein argonaute-2 (encoded by *AGO2*) has been identified as one such additional target, and, as this protein plays important roles in microRNA (miRNA) maturation and function [12], these data indicate that miRNAs also play a role in modulating the IMiD-induced cell death in MM [11].

More recently, long non-coding RNAs (lncRNAs) and circular RNAs (circRNAs) have also been implicated in MM pathogenesis [13,14,15]. However, data on circRNA expression in MM are scarce and these molecules have not previously been studied in relation to IMiD therapy.

CircRNAs are endogenously expressed single-stranded RNA molecules that are characterized by a covalently closed structure and a high stability within cells [16,17]. More circRNAs than the number of genes in the human genome have now been annotated and most are derived from protein coding genes [18,19]. Particular circRNAs are highly expressed, but often in a tissue-specific manner [20,21,22]. The biological functions are unknown for most circRNAs, but an increasing number of reports suggest that some circRNAs can inhibit the function of other molecules, including miRNAs and proteins, by sequestering them in the cytoplasm [23,24,25]. In particular, a circular RNA derived from a locus on chromosome X, which was named circular RNA sponging miR-7 (ciRS-7) due to its many inherent miR-7 binding sites [23], has been thoroughly studied, and functions through binding of miR-7 in the brain [23,26,27]. Recently, circRNAs have been implicated in normal cellular differentiation [28,29], but also in disease [30,31].

To expand our knowledge on the mechanisms of action of the IMiDs in MM and to identify novel molecules with potential as biomarkers, we characterized genome-wide circRNA expression changes upon in vitro development of IMiD resistance. This was done using high-throughput RNA-sequencing (RNA-seq) performed on ribosomal RNA (rRNA)-depleted total RNA samples from IMiD-sensitive MM cells and cells that developed resistance to lenalidomide and pomalidomide respectively, through prolonged treatment with low doses of these drugs [32]. We further investigated the most downregulated circRNA in both lenalidomide- and pomalidomide-resistant cells, ciRS-7, and found that it became epigenetically silenced in resistant cells and partially restored upon treatment with specific epi-drugs. However, ciRS-7 is unlikely to play a direct role in development of IMiD resistance since knockdown of ciRS-7 did not, in itself, affect cellular resistance to the drugs. 

## 2. Results

### 2.1. Genome-Wide circRNA Expression Patterns Reflect Sensitivity to IMiDs

We performed RNA-seq of rRNA-depleted total RNA from IMiD-sensitive cells (the NCI-H929 cell line, denoted as parental cells) and lenalidomide-resistant (LR) and pomalidomide-resistant (PR) counterparts, as well as of LR and PR cells treated with various epi-drugs. The RNA-seq data were analyzed for circRNA expression using a stringent version of the bioinformatics algorithm, find_circ [16]. In total, we detected 6368 circRNA candidates supported by at least two head-to-tail junction-spanning reads. Within each sample type, between 1044 and 2136 unique circRNAs were detected, of which between 214 and 420 were defined as high-abundance circRNAs (reads per million, RPM > 0.2) (Figure 1A). See Appendix A for full list of high-abundance circRNAs in all 11 samples.

We observed that the samples could be separated into two distinct groups when performing principal component analysis (PCA) using the top 200 most abundant circRNAs across all samples (Figure 1B) and when using all data (Appendix A). One group consisted of all samples from IMiD-sensitive cells (parental cells and LR and PR cells treated with an Enhancer of zeste homolog 2 inhibitor (EZH2i) alone or EZH2i in combination with a DNA methyl transferase inhibitor (DNMTi)), and the other group consisted of all samples from IMiD-resistant cells (untreated LR and PR cells as well as LR cells and PR cells treated with a histone deacetylase inhibitor (HDACi) or DNMTi alone). When including all data, the resistant group was further subdivided into two groups according to whether the cells were LR or PR (Appendix A). The PCA were corroborated by unsupervised hierarchical cluster analyses in which sample clustering also reflected sensitivity to the IMiDs (Figure 1C and Appendix A).

### 2.2. circRNA Expression Changes upon Acquired Resistance to IMiDs

Among the high-abundance circRNAs (Figure 1A), there was a large overlap between the circRNAs detected in the IMiD-sensitive cells and in the LR and PR counterparts and, in total, 393 unique high-abundance circRNAs were detected (Appendix A). Many of these circRNAs were up- or down-regulated upon acquired resistance (Figure 2A,B). Several circRNAs, which have previously been implicated in cancer, were upregulated both in the LR and in the PR cells, most noticeably circZKSCAN1 [33,34] and circCDYL [35]. Interestingly, we also observed upregulated circRNAs from the MM-related genes *IKZF3* and *WHSC1* (also known as *MMSET*), which are overexpressed as a result of the t(4;14) translocation in MM patients [36]. Among the circRNAs, which were downregulated both in LR cells and in PR cells, ciRS-7 [37,38] stood out as the most downregulated circRNA in both cell lines.

There was a large overlap in the specific circRNAs being upregulated upon acquired lenalidomide and pomalidomide resistance, respectively. The same was observed for downregulated circRNAs, whereas only very few circRNAs were downregulated upon acquired lenalidomide resistance and upregulated upon acquired pomalidomide resistance, and vice versa (Appendix A). This was expected as these two drugs have similar chemical structures and mechanisms of action [7,8,9] and because the PR cells are also LR [32]. More circRNAs were upregulated than downregulated upon acquired resistance (Figure 2A,B). This was most pronounced for PR cells, which was reflected by an overall significant increase in circRNA abundance (Appendix A). Interestingly, the observed changes in circRNA expression were largely independent of changes in host gene expression as a poor correlation between changes in linear expression and circRNA expression was observed, both when considering LR cells and PR cells (Figure 2C,D).

### 2.3. circRNA Expression Changes upon Combined DNMT and EZH2 Inhibition

Next, we investigated the circRNA transcriptomic changes in the LR and PR cells treated with an EZH2 inhibitor (EPZ-6438) in combination with a DNMT inhibitor (5-azacytidine). We detected 1513 and 1669 unique circRNA candidates, of which 261 and 312 had an RPM > 0.2 in the LR cells and PR cells treated with 5-azacytidine and EPZ-6438, respectively. Among the high-abundance circRNAs, there was a relatively large overlap between the circRNAs detected in these four cell lines and in total, 517 unique circRNAs were detected (Appendix A). Several circRNAs were up- and down-regulated respectively, upon combined DNMT and EZH2 Inhibition (Figure 3A,B). Most noticeably, circCDYL and circZKSCAN1, which were upregulated upon acquired resistance to lenalidomide, were downregulated in the LR cells treated with EZH2i and DNMTi (Figure 3A). Likewise, circHIPK3, which was downregulated in the LR cells, was upregulated upon the treatment (Figure 3A). Again, the observed changes in circRNA expression were largely independent of changes in host gene expression as a poor correlation between changes in linear expression and circRNA expression was observed, both when considering LR cells and PR cells (Figure 3C,D).

There was no tendency for circRNAs to be upregulated in both LR and PR cells treated with epi-drugs nor for circRNAs to be downregulated in both LR and PR cells treated with epi-drugs (Appendix A). Likewise, there was no significant overall increase or decrease in circRNA abundance upon treatment with epi-drugs in these cell lines (Appendix A).

### 2.4. Validation of circRNA Candidates from the RNA-Seq Data by NanoString nCounter Technology

Because ciRS-7 was the most downregulated circRNA in both LR and PR cells, and because circIKZF3 was upregulated in both LR and PR cells, we decided to validate the pattern of down- or up-regulation of these two circRNAs by NanoString nCounter^®^ technology, which has been described as a sensitive and quantitatively accurate methodology [39,40]. We did indeed observe the same pattern of downregulation of ciRS-7 (Figure 4A) and upregulation of circIKZF3 (Figure 4B) in resistant cells. Interestingly, ciRS-7 was not found to be upregulated in the RNA-Seq data upon treatment with epi-drugs, contrary to the NanoString data (Figure 4A). We hypothesize that this discrepancy can be explained by the fact that subtle expression changes might not be detected using RNA-seq, while the more sensitive NanoString method is able to detect even very small changes in expression of circRNAs. Furthermore, it has previously been shown that expression levels of circRNAs take time to build up in cells due to the relatively inefficient back-splicing reaction [41] and that circRNAs may not reach steady-state levels in fast dividing cells [42]. Together, this may explain the relatively subtle upregulation of ciRS-7 observed.

Since we were able to validate the pattern of down- and up-regulation of ciRS-7 and circIKZF3, we wanted to investigate if these two circRNAs were directly involved in acquired resistance. We began to examine this by knocking down ciRS-7 in the IMiD-sensitive cells, and circIKZF3 in LR and PR cells, respectively. Knockdown of ciRS-7 was performed using a previously established protocol based on small internally segmented interfering RNAs (sisiRNAs) [43]. While we achieved successful knockdown of ciRS-7 in IMiD-sensitive cells (Figure 4C), we were unable to knock down circIKZF3 in LR cells (Figure 4D) using two different siRNAs both targeting the back-splice junction of circIKZF3 (Appendix A), and we therefore decided to continue functional assays with ciRS-7 only. 

### 2.5. Downregulation of ciRS-7 Is Associated with Increased DNA Methylation Levels

It has previously been described that ciRS-7 is produced as part of an upstream long non-coding RNA (LINC00632) [38,44], which encodes three different transcripts denoted as T1, T2 and T3 [44] (Figure 5A). Thus, we assessed the expression levels of each of these transcripts using reverse transcriptase-quantitative PCR (RT-qPCR) in the IMiD-sensitive cells, the LR cells and the PR cells, and found that only T3 was expressed (Figure 5B). Moreover, as observed for ciRS-7, we found that T3 was much higher expressed in the IMiD-sensitive cells and higher expressed in LR cells compared to PR cells (Figure 5B). Interestingly, the promoter region of the T3 transcript is associated with a CpG island (Figure 5A). Thus, we assessed the methylation levels of this CpG island using Sensitive Melting Analysis after Real Time-Methylation-Specific PCR (SMART-MSP) [45] and found an increased methylation of more than 10-fold in LR cells, and more than 20-fold in PR cells, relative to the IMiD-sensitive cells (Figure 5C). These data were supported by semi-quantitative bisulfite sequencing data, as we found that most CpG sites were completely unmethylated in the IMiD-sensitive cells (Figure 5D), whereas relatively higher methylation levels were observed in LR cells (Figure 5E) and we observed the highest levels in PR cells (Figure 5F). Together, these analyses indicate that ciRS-7 become epigenetically silenced during acquired resistance to IMiDs in multiple myeloma cells.

### 2.6. Combined DNMT and EZH2 Inhibition Reactivates ciRS-7 through T3

Since we found that ciRS-7 was epigenetically silenced during acquired resistance to IMiDs, we assessed the expression changes of the T3 transcript of LINC00632 (Figure 5A) upon treatment of LR and PR cells with 5-azacytidine and EPZ-6438. Indeed, we observed a strong upregulation of T3 upon the treatment with epi-drugs in both the LR and the PR cells (Figure 6). Similar to the stronger upregulation of ciRS-7 observed in the LR cells compared to the PR cells in the NanoString data (Figure 4A), the upregulation of the T3 transcription was also most pronounced in the LR cells.

### 2.7. Knockdown of ciRS-7 Does Not Make the IMiD-Sensitive Cells Become More Resistant

Next, we investigated whether ciRS-7 could be directly involved in mediating IMiD resistance in MM cells. Since ciRS-7 harbor many miRNA binding sites (Figure 7A), we first performed a comprehensive profiling of the expression levels of 799 miRNAs using NanoString nCounter technology within the 11 samples used in this study (Figure 1A). In particular, ciRS-7 contains many miR-7 binding sites and is, therefore, most likely to function through binding of miR-7 molecules, however, miR-7 expression did not correlate with the expression of ciRS-7 (Figure 7B). On the other hand, the expression pattern of miR-1290, which has been implicated in mediating drug resistance in other cancers [46,47], followed the same expression pattern as ciRS-7 (Figure 7B). Therefore, we performed a knockdown of ciRS-7 in the IMiD-sensitive cells (Figure 7C) and investigated if this affects drug sensitivity. However, we found no difference in drug sensitivity between the ciRS-7 knockdown and control cells (Figure 7D,E), implying that changes in ciRS-7 expression may not be the cause of resistance.

### 2.8. ciRS-7 Expression in MM Cell Lines Does Not Reflect IMiD Sensitivity 

Next, we profiled ciRS-7 expression in several independent MM cell lines (JJN3, RPMI-8826, LP1, MOLP2, MOLP8, EJM and OPM2) with different primary IMiD sensitivity/resistance. The JJN3, RPMI-8826 and LP1 cell lines are resistant, whereas MOLP2, MOLP8 and EJM are partially resistant and OPM2 and NCI-H929 are IMiD-sensitive. As could be expected from the previously described results, we found no correlation between ciRS-7 expression level and IMiD sensitivity (Appendix A). Notably, ciRS-7 is completely absent in OPM2, which was the only other sensitive cell line than NCI-H929, and highly expressed in the partially resistant EJM and MOLP8 cell lines.

### 2.9. ciRS-7 Becomes Epigenetically Silenced after Prolonged Cell Culture without Drug-Exposure

Because we did not observe any effect of ciRS-7 knockdown on IMiD sensitivity, and because we did not see a correlation between ciRS-7 expression and primary IMiD sensitivity in a collection of MM cell lines, we decided to investigate if ciRS-7 may become epigenetically silenced as a result of prolonged cell culturing even in the absence of the drugs. Because of the pronounced downregulation observed in both LR and PR cells, we were surprised to find that ciRS-7 expression was also diminished because of prolonged culturing of the cells (Appendix A). The downregulation of ciRS-7 was again found to correlate with increased promoter methylation levels (Appendix A). To investigate if this may also apply to other MM cell lines, we performed the same experiment using MOLP8 cells, which express high levels of ciRS-7 (Appendix A). Interestingly, we found that ciRS-7 was not silenced because of prolonged culturing of these cells (Appendix A), despite the fact that epigenetic silencing of the T3 transcript occurred (Appendix A). However, this might be explained by the T2 transcript of LINC00632 (Figure 5A) being the main driver of ciRS-7 expression in MOLP8 cells (Appendix A).

## 3. Discussion

Acquired resistance to IMiDs is a major challenge in the clinical management of MM patients. Because circRNAs have recently been implicated in cancer and show promise as biomarkers [31], we profiled genome-wide circRNA expression patterns in a previously established in vitro model of IMiD resistance in MM [32], in a search for potential novel mediators of acquired IMiD resistance. Interestingly, individual samples clustered according to IMiD sensitivity and we found several circRNAs, which have previously been implicated in cancer, to be either upregulated or downregulated upon acquired IMiD resistance. From these analyses, ciRS-7 emerged as the best candidate for being a novel mediator of IMiD resistance as it was the most downregulated circRNA upon acquired resistance to both lenalidomide and pomalidomide. In addition, ciRS-7 has been implicated in many different cancers [31,38] and does not have a protein coding host gene. NanoString nCounter analysis [39] confirmed the abrupt downregulation observed in the RNA-seq data: it was downregulated by approximately 20- and 200-fold in lenalidomide- and pomalidomide-resistant cells, respectively. By RT-qPCR analyses of the three different transcripts (T1, T2 and T3) of the long non-coding RNA locus in which ciRS-7 is embedded [44], we found that expression of the T3 transcript correlated with expression of ciRS-7. Because this transcript has a CpG island in its promoter region, we hypothesized that ciRS-7 was epigenetically silenced upon acquired resistance. Indeed, we found that this promoter CpG island was unmethylated in the IMiD-sensitive cells, but became hypermethylated in the resistant cells. This was most pronounced in the pomalidomide-resistant cells, which also showed the lowest expression of both the T3 transcript and of ciRS-7. Finally, we were able to show that the T3 transcript, as well as ciRS-7, could be re-activated upon treatment with epi-drugs, including a DNMT inhibitor, thus providing additional evidence that ciRS-7 is epigenetically regulated through methylation of this CpG island.

Because we have previously shown that acquired IMiD resistance is, in part, an epigenetic phenomenon and that resistant cells can be re-sensitized by treatment with epi-drugs [32], we wanted to investigate if ciRS-7 could be directly involved in mediating IMiD resistance. Because ciRS-7 harbors many miRNA binding sites, in addition to the miR-7 binding sites, we analyzed the expression levels of miRNAs for which more than five binding sites are present in the ciRS-7 sequence as well as miR-671, which can facilitate cleavage of ciRS-7 through one highly complementary site [49]. While miR-7 was generally lowly expressed and did not correlate well with the expression patterns of ciRS-7, we found that the expression levels of miR-1290 and ciRS-7 were correlated. We have recently shown that ciRS-7 does not co-localize with miR-7 in several solid tumors [50] and another study found that it may function independent of miR-7 in malignant melanoma [38]. miR-1290 has not previously been implicated in MM or IMiD sensitivity to our knowledge but has been shown to mediate drug resistance in other cancers [46,47]. Thus, we decided to perform a knockdown of ciRS-7 in the IMiD-sensitive cells and analyze their response to IMiDs after five days. Depleting the cells of ciRS-7 did not, however, make the cells more resistant to lenalidomide. This disappointing result was observed both when analyzing proliferation and apoptosis rates upon treatment with the drug. Therefore, we investigated the possibility that ciRS-7 becomes epigenetically silenced as a result of prolonged culturing of the cells even without drug-exposure. Indeed, we found that ciRS-7 expression was completely diminished when the cells were cultured for longer periods and this correlated with increased methylation levels of the T3 promoter CpG island. Together, these data suggest that ciRS-7 is not directly responsible for the acquired IMiD resistance. On the other hand, we speculated that cells in which ciRS-7 becomes epigenetically silenced may have a selective advantage and therefore become the dominant clone in the cell culture. This hypothesis is in line with the study by Hanniford and co-workers, showing that ciRS-7 can function independent of miR-7 and have tumor suppressor properties in malignant melanoma [38]. Thus, we analyzed another MM cell line, MOLP8, in which ciRS-7 is abundantly expressed. While ciRS-7 was also dynamically expressed in MOLP8 cells, it was not silenced as a result of prolonged culturing and we found that the T2 transcript was expressed in MOLP8 and correlated with expression levels of ciRS-7, whereas the T3 transcript, although initially expressed, became epigenetically silenced after about one month. The fact that the T3 transcript can be epigenetically silenced in cells that also express the T2 transcript, and therefore maintain high levels of ciRS-7, indicates that the increased methylation levels of the T3 promoter CpG island may be a passenger event and, thus, arguing against our hypothesis that ciRS-7 has an anti-proliferative effect in MM cells. However, in cells where expression of ciRS-7 is primarily driven by the T3 transcript, this mechanism of epigenetic shut-down of ciRS-7 is highly interesting and illustrates the importance of including an additional control consisting of cells being grown for the same duration as the drug treatment in future in vitro studies of acquired drug resistance. This control should be analyzed alongside the other samples to account for genetic or transcriptomic changes caused by prolonged growth in culture. While it seems that ciRS-7 is not directly involved in mediating IMiD resistance, we cannot exclude that other circRNAs may be. We were intrigued by the fact that the samples in our genome-wide circRNA profiling experiment clustered according to IMiD sensitivity and that most of the observed circRNA expression changes were independent of their cognate linear host genes. However, this cannot, by itself, be considered as evidence of circRNAs serving a direct role in acquired IMiD resistance in MM, as cellular proliferation rates [42,51,52], epigenetic changes [29,53] and factors functioning in trans [54,55,56] may modulate the overall expression levels of circRNAs. Future studies of acquired drug resistance should focus on investigating other circRNA candidates and preferably include paired primary patient samples from time of diagnosis and relapse following acquired drug resistance.

## 4. Materials and Methods

### 4.1. Cell Cultures and Treatments

The human MM cell lines NCI-H929, JJN3, RPMI-8826, LP1, MOLP2, MOLP8, EJM and OPM2 were grown as described in Reference [32], at 37 °C and 5% CO_2_. Cell density and viability were determined using the EVE automated cell counter (NanoEnTek). The IMiD-resistant NCI-H929 cells, PR cells and LR cells were previously established by continuous culture of NCI-H929 cells in the presence of low-dose lenalidomide or pomalidomide for 4–6 months [32]. The PR cells are also resistant towards lenalidomide. The PR and LR cells were treated with different combinations of epi-drugs: the DNMTi, 5-azacytidine, the HDACi, Panobinostat, the EZH2i, EPZ-6438, and a combination of DNMTi and EZH2i, as previously described [32]. All cell lines were acquired from Dimopoulos et al. [32].

### 4.2. RNA and DNA Isolation

Total DNA and RNA were isolated from the cell lines using the AllPrep DNA/RNA/miRNA Universal Kit (Qiagen, Hilden, Germany) according to the manufacturer’s protocol. The quantity of total DNA and RNA was measured by spectrophotometry using a NanoDrop spectrophotometer (Thermo Scientific, Waltham, MA, USA).

### 4.3. Library Preparation and High-Throughput RNA-seq

One microgram total RNA was rRNA depleted using the Ribo-Zero rRNA Removal Kit (Human, Mouse, Rat) (Epicentre Biotechnologies, Madison, WI, USA) according to the manufacturer’s instructions, followed by a purification step using AMPure XP Beads (Beckman Coulter, Brea, CA, USA). Sequencing libraries were generated using the ScriptSeq v2 RNA-Seq Library Preparation Kit (Epicentre) according to the manufacturer’s instructions using 12 PCR cycles. Purification was performed using AMPure XP Beads. The final libraries were quality controlled on the 2100 Bioanalyzer (Agilent Technologies, Santa Clara, CA, USA) and quantified using the KAPA library quantification kit (Kapa Biosystems, Wilmington, MA, USA) according to the manufacturer’s protocol. High-throughput sequencing was performed on the HiSeq 4000 system (Illumina, San Diego, CA, USA) at the Beijing Genomics Institute (BGI) in Copenhagen using the 100 paired-end sequencing protocol.

### 4.4. RNA-seq Data Analysis

Sequencing data were quality filtered (Phred score 20) and adapter trimmed using Trim Galore. Filtered and trimmed sequencing reads were mapped to the human genome (hg19) using TopHat2. CircRNA expression levels (sequencing reads aligning across particular back-splicing junctions) were quantified based on a stringent version of the find_circ bioinformatics algorithm [56], normalized to the total number of raw reads and communicated as RPM. Host gene expression levels were calculated as the average linear reads corresponding to the splice donor and splice acceptor sites of each circRNA in question, normalized to the total number of raw reads and communicated as RPM. Heatmaps, hierarchical cluster analyses and PCA plots were generated using R software version 3.5.1 with the following packages installed: ggplot2, ComplexHeatmap, circlize, dendextend and RColorBrewer. The PCA was done using log2 transformed data and the cluster analyses were performed using the Pearson distance calculation method and the ward.D clustering method after computing the z-score for each data point. Waterfall plots were generated using Excel 2016 software (Microsoft Corporation, Redmond, WA, USA) and scatter plots and violin plots were generated using Prism 8 software (GraphPad, La Jolla, CA, USA).

### 4.5. RT-qPCR Analyses of the LINC00632 T1, T2 and T3 Transcripts

One hundred to five hundred nanograms of RNA was reverse-transcribed using the SuperScript IV First-Strand Synthesis kit (Thermo Scientific) according to the manufacturer’s protocol. The samples were diluted 1:5 in nuclease-free water. RT-qPCR was performed using 4 μL of diluted cDNA and 6 μL of LightCycler^®^ 480 SYBR Green I Master (Roche Life Science, Mannheim, Germany). The PCR amplification was performed on the LightCycler 480 instrument II (Roche Life Science) with the following cycling conditions: one cycle of 95 °C for 10 min, followed by 40 cycles of 95 °C for 10 s, 60 °C for 10 s and 72 °C for 20 s. All RT-qPCR primers are shown in Appendix A. Reference genes *SF3A1* and *PUM1* were used to normalize the data and have previously been shown to be stable in MM [32]. The experiments were done as technical triplicates at the level of cDNA synthesis.

### 4.6. Sensitive Melting Analysis after Real-Time Methylation-Specific PCR (SMART-MSP)

Five hundred nanograms of genomic DNA for each sample were bisulfite-treated using the EpiTect Bisulfite kit (Qiagen) according to manufacturer’s protocol. SMART-MSP primers were designed to specifically amplify bisulfite-treated and methylated DNA by targeting several CpG sites and by placing the cytosine of a CpG site near or at the 3’ end of the primer (Appendix A). We used a previously published assay that target CpG-deprived Alu sequences [57] for normalization, as this assay is less susceptible to normalization errors caused by copy number changes and aneuploidy [58]. Bisulfite-converted fully methylated and fully unmethylated DNA (Qiagen) was used as positive and negative controls, respectively. The negative control was considered negative when amplification occurred after more than 35 PCR cycles. qPCR was performed using a 384-well plate with 2 µL of bisulfite-treated DNA and 8 µL of LightCycler^®^ 480 High-Resolution Meting Master (Roche Life Science) including primers, in each well. The PCR amplification was carried out with the following cycling conditions: one cycle of 95 °C for 10 min, followed by 45 cycles of 95 °C for 10 s, 60 °C for 20 s and 72 °C for 20 s. The melting program was carried out using the following conditions: 95 °C for 1 min, 40 °C for 1 min, and 20 acquisitions/°C from 65 °C to 90 °C. The PCR amplification was performed on a LightCycler 480 instrument II (Roche Life Science). The experiments were done as technical triplicates at the level of qPCR.

### 4.7. Bisulfite Sequencing of the LINC00632 T3 Promoter CpG Island

One microgram of genomic DNA for each sample was bisulfite-treated using the EpiTect Bisulfite kit (Qiagen) according to manufacturer’s protocol. The PCR amplification was carried out with the following cycling conditions: one cycle of 94 °C for 3 min, followed by 45 cycles of 94 °C for 20 s, 60 °C for 20 s and 72 °C for 30 s, followed by one cycle of 72 °C for 10 min. The PCR amplification was performed on a SimpliAmp Thermal Cycler (Thermo Scientific). PCR products were purified using GeneJET PCR Purification Kit (Thermo Scientific). Amplicons were Sanger sequenced in both forward and reverse direction using the service of GATC, Eurofins Genomics. Methylation-independent primers for amplification and sequencing are shown in Appendix A.

### 4.8. NanoString nCounter Analysis of ciRS-7 and circIKZF3

The NanoString nCounter analyses were performed using a previously published custom-designed panel, in which ciRS-7 and circIKZF3 were included [59]. One hundred nanograms of total RNA was used and the hybridization was carried out for 20 hours before performing nCounter SPRINT (NanoString Technologies, Seattle, WA, USA) analyses according to the manufacturer’s instructions. The raw data were processed using the nSOLVER 4.0 software (NanoString Technologies). Background subtraction and normalization using the geometric mean of all positive controls was performed as recommended by the manufacturer. Finally, a second normalization using the geometric mean of the four most stable linear reference genes with reasonable expression levels (*ACTB*, *ESD*, *SF3A1* and *PUM1*) was performed, before exporting the data to Excel (Microsoft Corporation).

### 4.9. NanoString nCounter Analysis of miRNAs

The nCounter Human v3 miRNA panel (NanoString Technologies), which target 799 miRNAs, was used for miRNA profiling. One hundred nanograms of total RNA from each sample was subjected to nCounter™ SPRINT (NanoString Technologies) analysis according to the manufacturer’s instructions. The raw data were processed using the nSOLVER 3.0 software (NanoString Technologies). Background subtraction and normalization using the geometric mean of all positive controls was performed as recommended by the manufacturer. A second normalization was performed using the geometric mean of the top 100 highest expressed miRNAs, before exporting the data to Excel (Microsoft Corporation).

### 4.10. ciRS-7 and circIKZF3 Knockdown

siRNA transfections were performed in 6-well plates using 1.5 mL media per well using Lipofectamine RNAiMAX (Thermo Scientific) according to the manufacturer’s protocol. Knockdown of ciRS-7 in sensitive NCI-H929 cells was performed using two different sisiRNAs [43], one sisiRNA targeting the back-splice junction of ciRS-7 (BSJ sisiRNA), and the other targeting an internal sequence of ciRS-7 (Internal sisiRNA). For knockdown of circIKZF3, two different siRNAs both targeting the back-splice junction of circIKZF3 were used. The sisiRNA and siRNA sequences can be found in Appendix A. MISSION^®^ siRNA Universal Negative Control #2 (Sigma, St. Louis, MO, USA) was used as a negative control. Cells were incubated with sisiRNAs or siRNAs for 48 hours with no media change.

### 4.11. Proliferation Assay 

Ten thousand sisiRNA-transfected cells were seeded in 100 μL of media in 96-well plates. Lenalidomide was added to separate wells in order to generate wells with eight different concentrations (100, 50, 10, 5, 1, 0.1, 0.01 and 0.001 μM) of each of the drugs. Seventy-two hours after drug addition, the proliferation assay was performed using the TACS XTT Cell Proliferation Kit (R & D Systems, Minneapolis, MN, USA) according to the manufacturer’s protocol. The XTT reagent was added and after three hours of incubation, the cells were analyzed spectrophotometrically on a CLARIOstar microplate reader by measuring absorbance at 490 nm, with a reference wavelength of 630 nm.

### 4.12. Apoptosis Assay

The apoptosis assay was performed using the Annexin V-FITC Apoptosis Detection Kit (Sigma) according to the manufacturer’s protocol. The samples were analyzed using a CytoFLEX Flow Cytometer (Beckman Coulter). A positive control was created by inducing apoptosis in parental NCI-H929 cells by incubating them for one hour with 1 μg/mL staurosporine. Untreated parental NCI-H929 cells were used as negative control. A compensation matrix was applied to the data to account for spectral overlap between fluorophores, by measuring the amount of fluorescence from unlabeled control cells, PI-only control cells and FITC-only control cells. The Annexin V-FITC conjugate binds to translocated phosphatidylserines. The intensity of the FITC signal, therefore, corresponds to the level of early apoptosis. Propidium Iodide (PI) labels DNA in cells where the membrane has been completely compromised. The intensity of the PI signal, therefore, corresponds to the level of dead cells.

### 4.13. Statistical Analyses

All statistical analyses were performed using Prism 8.2.0 (GraphPad, La Jolla, CA, USA). Linear regression was used to assess potential correlations between changes in circRNA expression and expression changes in their cognate linear host genes. Linear regression was also used to assess potential correlations between circRNA expression changes upon acquired lenalidomide resistance and pomalidomide resistance as well as to assess changes upon treatment of LR cells and PR cells with 5-azacytidine and EPZ-6438. Mann–Whitney tests were used to assess potential differences between overall circRNA expression levels in IMiD-sensitive cells versus LR cells and PR cells respectively, as well as to assess potential differences between overall circRNA expression levels in LR cells and PR cells upon treatment with 5-azacytidine and EPZ-6438. Unpaired *t*-tests were used to assess potential differences in ciRS-7 expression and expression of the T3 transcript of LINC00632 and its methylation level respectively, in IMiD-sensitive cells relative to LR cells and PR cells, as well as to assess potential differences in expression of the T3 transcript of LINC00632 in LR cells and PR cells upon treatment with 5-azacytidine and EPZ-6438. All *p*-values were two-tailed and considered significant if <0.05.

## 5. Conclusions

We found that genome-wide circRNA expression patterns are affected by IMiD sensitivity in an in vitro model of acquired drug resistance. Most of the observed circRNA expression changes, upon acquired IMiD resistance, occurred independent of changes in their cognate linear host genes. While we found ciRS-7 to be epigenetically silenced upon acquired resistance to both lenalidomide and pomalidomide, it does not seem to be directly involved in mediating resistance to these drugs. Future studies should address the potential roles of other circRNAs in IMiD resistance and aim to uncover if particular circRNAs may have value as predictive biomarkers in a clinical setting.

## Figures and Tables

**Figure 1 cancers-13-00365-f001:**
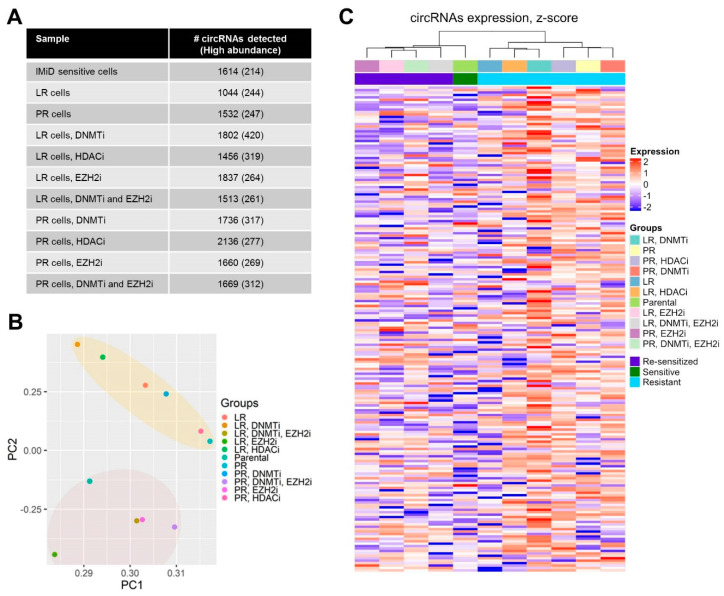
Genome-wide circRNA expression patterns reflect IMiD sensitivity in multiple myeloma cell lines. (**A**) Number of circRNAs (total and high abundance) detected in each of the 11 samples used in the study. (**B**) Principal component analysis using the 200 highest expressed circRNAs within the entire dataset. Grouping of samples according to IMiD sensitivity is highlighted. (**C**) Heatmap and hierarchical cluster analysis showing expression and clustering of the 200 highest expressed circRNAs (rows) within the entire data set. LR, lenalidomide-resistant, PR, pomalidomide-resistant.

**Figure 2 cancers-13-00365-f002:**
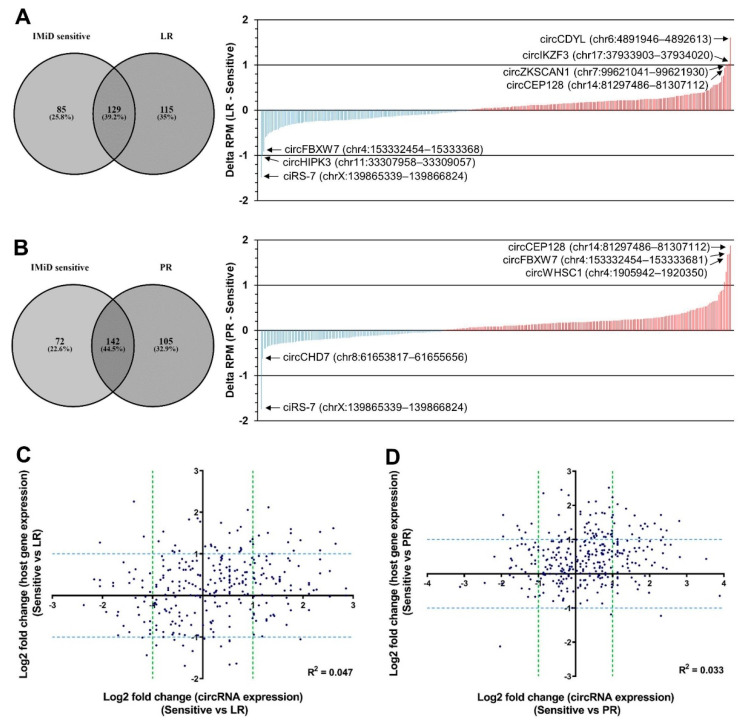
Changes in circRNA expression upon acquired resistance to IMiDs. (**A**,**B**) Waterfall plots illustrating the upregulated (in red) and downregulated circRNAs (in blue) in LR cells (**A**) and PR cells (**B**) relative to parental (IMiD-sensitive) cells. CircRNAs defined as high abundance in either parental cells, LR cells or PR cells, as shown in the Venn diagrams (left), were included in the analysis. (**C**,**D**) Scatter plots of log2 fold change in circRNA expression and log2 fold change in host gene expression in LR (**C**) and PR (**D**) cells compared to parental (sensitive) cells. Potential correlations were assessed by calculating Pearson correlation coefficients, R^2^. LR, lenalidomide-resistant, PR, pomalidomide-resistant.

**Figure 3 cancers-13-00365-f003:**
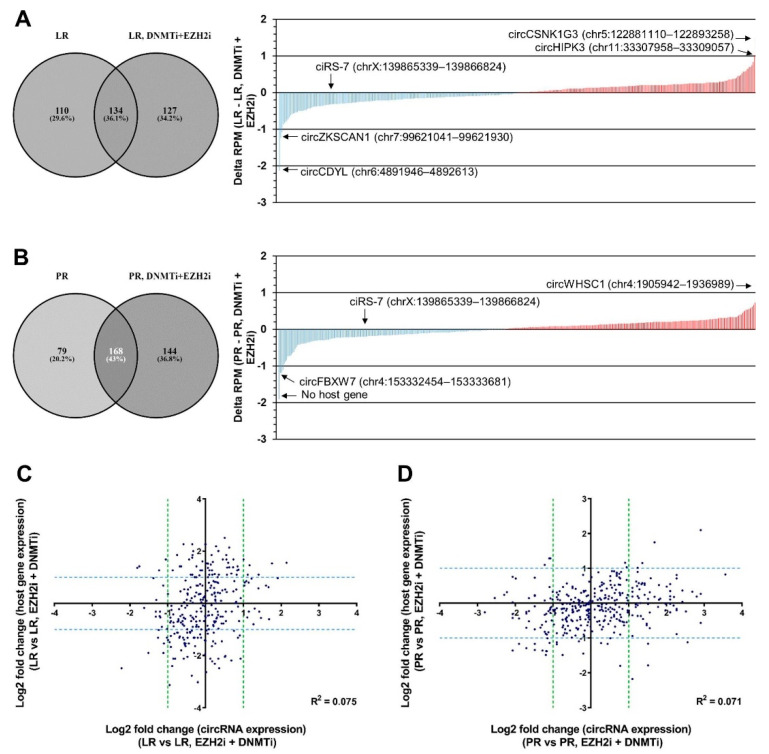
circRNA expression changes upon combined DNMT and EZH2 inhibition. (**A**,**B**) Waterfall plots illustrating the upregulated (in red) and downregulated (in blue) circRNAs in LR cells treated with DNMTi and EZH2i (**A**) and PR cells treated with DNMTi and EZH2i (**B**) relative to LR cells and PR cells, respectively. CircRNAs defined as high abundance in either LR cells, LR cells treated with DNMTi and EZH2i, PR cells, or PR cells treated with DNMTi and EZH2i, as shown in the Venn diagrams (left), were included in the analysis. (**C**,**D**) Scatter plots of log2 fold change in circRNA expression and log2 fold change in host gene expression in LR cells treated with DNMTi and EZH2i (**C**) and PR cells treated with DNMTi and EZH2i (**D**) compared to LR cells and PR cells, respectively. Potential correlations were assessed by calculating Pearson correlation coefficients, R^2^. LR, lenalidomide-resistant, PR, pomalidomide-resistant.

**Figure 4 cancers-13-00365-f004:**
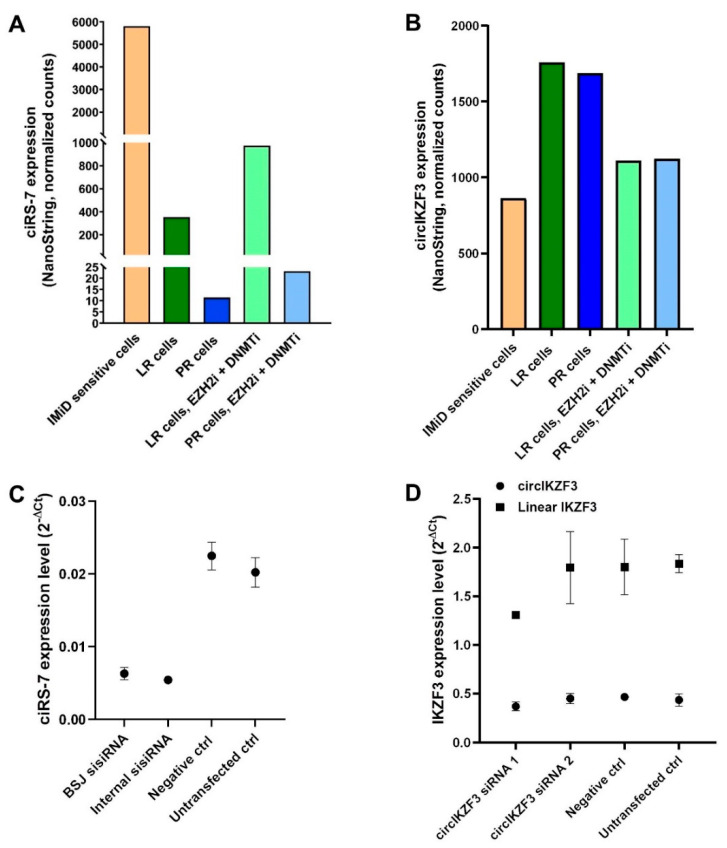
Validation of circRNA candidates from the RNA-Seq data by NanoString nCounter technology (**A**,**B**) ciRS-7 expression (**A**) and circIKZF3 expression (**B**) (analyzed by NanoString nCounter technology) in IMiD-sensitive, LR cells, PR cells, LR cells treated with EZH2i and DNMTi, and PR cells treated with EZH2i and DNMTi. (**C**) Expression of ciRS-7 five days post sisiRNA transfection. Relative expression of ciRS-7 (analyzed by reverse transcriptase-quantitative PCR (RT-qPCR)) in cells transfected with two different sisiRNAs targeting ciRS-7 (denoted as BSJ and Internal). Error bars reflect technical triplicates. (**D**) Expression of circIKZF3 five days post siRNA transfection. Relative expression of circIKZF3 (analyzed by RT-qPCR) in cells transfected with two different siRNAs targeting the back-splice junction of circIKZF3. Error bars reflect technical triplicates. BSJ: back-splicing junction, LR, lenalidomide-resistant, PR, pomalidomide-resistant.

**Figure 5 cancers-13-00365-f005:**
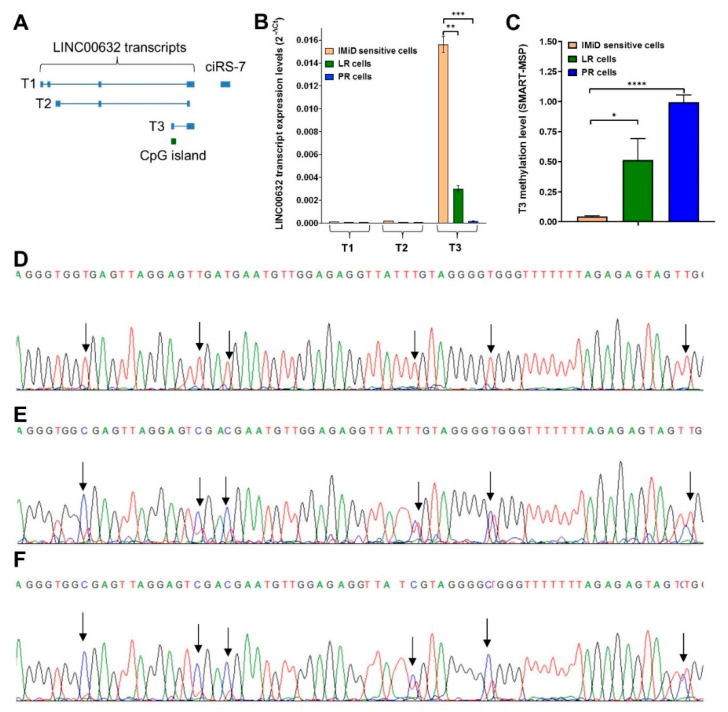
Downregulation of ciRS-7 is associated with increased methylation of the promoter driving its expression. (**A**) Schematic representation the three transcript variants, denoted as T1, T2 and T3, of LINC00632. The CpG island associated with T3 is indicated (not drawn to scale). (**B**) Expression levels (analyzed by RT-qPCR) of the LINC00632 transcript variants in parental (IMiD-sensitive), LR and PR cells. Error bars reflect technical triplicates. (**C**) Relative methylation levels (analyzed by SMART-MSP) of T3 in IMiD-sensitive, LR and PR cells, normalized to a fully methylated control. Error bars reflect technical triplicates. (**D**–**F**) Sanger sequencing chromatograms of T3 in parental (sensitive) cells (**D**), LR cells (**E**) and PR cells (**F**), with arrows indicating the CpG sites. LR, lenalidomide-resistant, PR, pomalidomide-resistant. **p* < 0.05, ***p* < 0.01, ****p* < 0.001, *****p* < 0.0001 (unpaired *t*-test).

**Figure 6 cancers-13-00365-f006:**
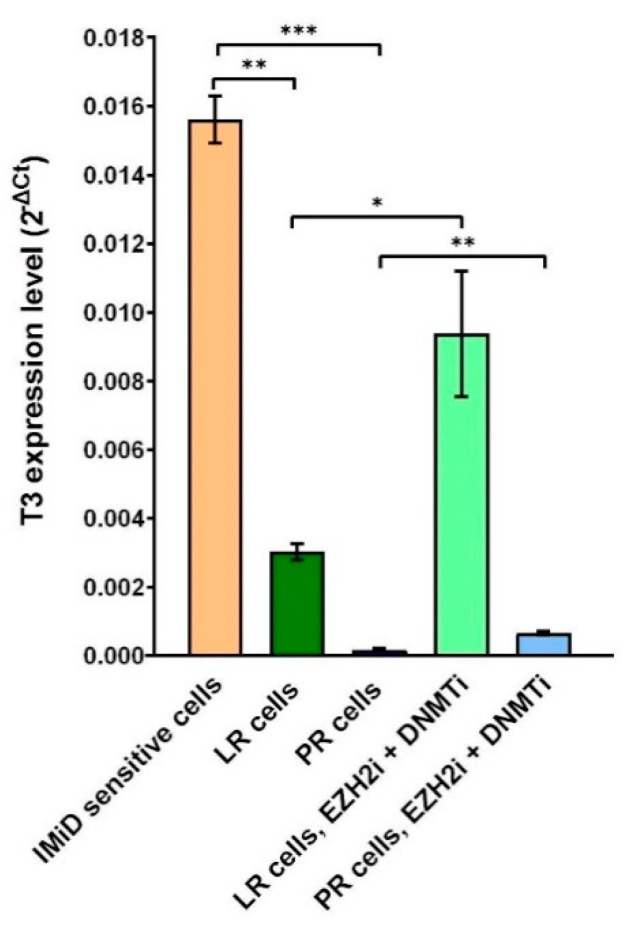
Combined DNMT and EZH2 inhibition reactivates ciRS-7 expression through T3. Relative expression (analyzed by RT-qPCR) of T3 in IMiD-sensitive cells, LR cells, PR cells, LR cells treated with EZH2i and DNMTi, and PR cells treated with EZH2i and DNMTi. LR, lenalidomide-resistant, PR, pomalidomide-resistant. **p* < 0.05, ***p* < 0.01, ****p* < 0.001 (unpaired *t*-test).

**Figure 7 cancers-13-00365-f007:**
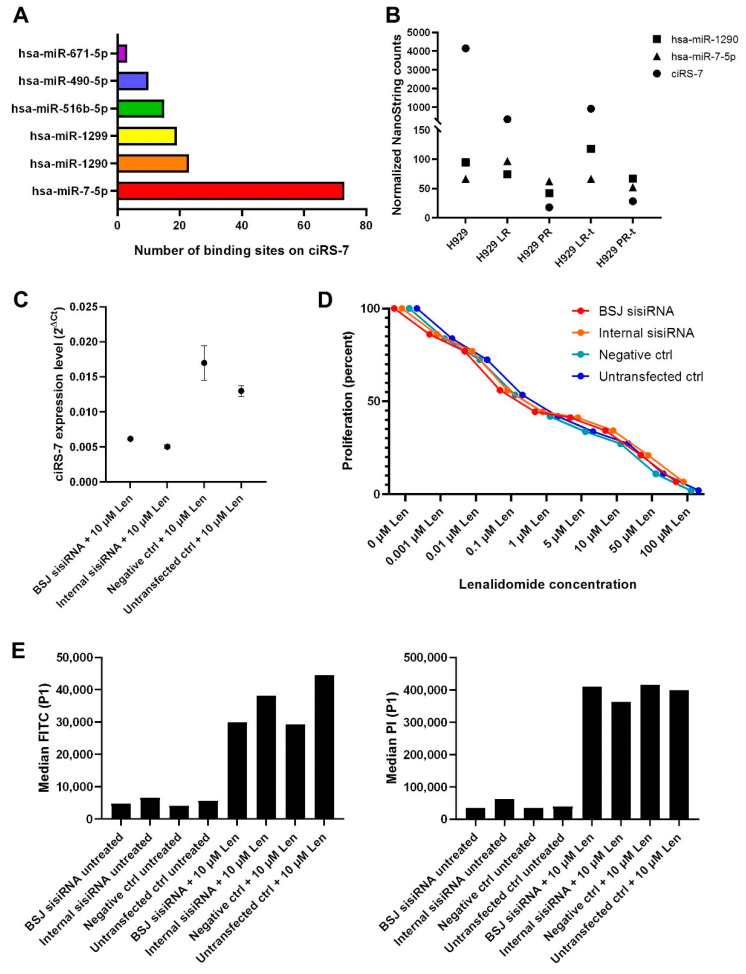
Knockdown of ciRS-7 does not induce IMiD resistance in sensitive cells. (**A**) The top five miRNAs with binding sites on ciRS-7, according to CircInteractome [48], and miR-671. (**B**) miR-7, miR-1290 and ciRS-7 expression in IMiD-sensitive (H929), LR cells, PR cells, LR cells treated with EZH2i and DNMTi (LR-t), and PR cells treated with EZH2i and DNMTi (PR-t). (**C**) Expression of ciRS-7 five days post sisiRNA transfection. Relative expression of ciRS-7 (analyzed by RT-qPCR) in cells transfected with two different sisiRNA targeting ciRS-7 (denoted as BSJ and Internal) treated with 10 μM lenalidomide. For ciRS-7 expression after knockdown in untreated cells, see Figure 4C. Error bars reflect technical triplicates. (**D**) Dose-response curves for lenalidomide in sisiRNA-transfected IMiD-sensitive cells. (**E**) Measurement of early apoptotic cells using fluorescein isothiocyanate (FITC) signal, and necrotic cells using propidium iodide (PI) signal, from sisiRNA-transfected parental cells treated with 10 μM lenalidomide. LR: lenalidomide-resistant, PR: pomalidomide-resistant, BSJ: back-splicing junction, ctrl: control, Len: lenalidomide.

## Data Availability

The data presented in this study are openly available in the Gene Expression Omnibus (GEO) database under accession number GSE161960. See Appendix A for a full list of high-abundance circRNAs, including chromosomal location, CTL ratio and RPM in all 11 samples.

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
