# Peer review of "Genome-Wide Circular RNA Expression Patterns Reflect Resistance to Immunomodulatory Drugs in Multiple Myeloma Cells"

_cancers, 2021, doi:10.3390/cancers13030365_

Round 1

Reviewer 1 Report

Authors have given a response to some of my comments (for example they explain and justify the reason why it is not applicable for circHIPK3 to perform the same epigenetic studies carried out with ciRS-7).

However, most of my concerns have not been addressed yet by the authors (e.g. functional studies with circCDYL, circZKSCAN1 and circHIPK3, experiments with resistant cells derived from other MM cell lines, etc). I understand time constrictions for some of them but, from my point of view, at least some of these issues should be addressed so that the work is not simply descriptive and has a real impact on understanding the mechanisms of resistance to IMiDs.

Author Response

We thank the reviewer for the constructive comments. However, we do not agree that our work is simply descriptive, as we have done functional studies for selected circRNAs and investigated underlying epigenetic changes behind the abrupt changes in ciRS-7 expression that we observed. We focused mostly on ciRS-7 as it was the most downregulated circRNA in both lenalidomide and pomalidomide resistant cells. While the data with respect to IMiD sensitivity for ciRS-7 turned out to be negative, we did, for the first time show that ciRS-7 can be epigenetically regulated through methylation of an upstream promoter CpG island. These data will be of interest to the many researchers working on ciRS-7 in particular and on circular RNAs in general. While we agree that it would be of interest to study additional circRNAs functionally, this is not feasible within the timeline for this paper, as we got 10 days for this second revision and as we are currently under lockdown and do not have cells growing. In addition, the cells that we do have in frozen aliquots have now been passaged too many times, and it would therefore not be meaningful nor reproducible to perform additional experiments without generating more IMiD resistant cell lines. However, this may take more than a year considering the time needed to identify the right concentration that will make the cells resistant over time without killing them.

In the first revision, we did address several of this reviewers comments, but frankly the suggested experiments would lead to generation of 2-3 times the data that was included in the first draft. Therefore, due to time and funding constraints, it is simply not possible to carry out all of that work. We believe in the importance of publishing meaningful results in a timely manner, so that other researchers in the field can benefit from this, rather than keeping the results to ourselves to perform functional studies for all of the very interesting candidate circRNAs that we identified by ourselves.

Reviewer 2 Report

Please confirm that the follow statement from the figure 4 legend is correct

"Error bars reflect technical triplicates".

Figure 4 and 7 a-please present standard deviation.

It it is correct additional validation of biological replicates need to be performed to perform the validation of the experimental data.

Author Response

We confirm that in figure 4C and 4D, the error bars reflect technical triplicates. Figure 4A and 4B represent normalized NanoString counts of ciRS-7 and circIKZF3, respectively. Thanks to the NanoString nCounter technology, the reproducibility is much higher compared to e.g. qPCR, and it is therefore not common to run technical replicates when using this technology. Furthermore, the company (NanoString Technologies) does not recommend running technical replicates (see “https://www.nanostring.com/support/product-support/knowledge-basefaqs”).

Figure 7A shows number of binding sites of selected miRNAs on ciRS-7 and standard deviation is therefore not applicable. Perhaps the reviewer was thinking of Figure 7B, which shows normalized NanoString counts of miR-1290, miR-7 and ciRS-7? In that case, please see answer above.

In this study, we performed an exploratory investigation of circRNAs in lenalidomide resistant and pomalidomide resistant cell lines and compared with their sensitive counterparts including both the parental cells as well as cells that were re-sensitized by treatment with various combinations of epigenetic drugs. We found several candidate circRNAs and validated two of these using NanoString nCounter technology. Additional candidates will be further studied in future studies.

Reviewer 3 Report

The authors commented well on the raised concerns. unfortunately, the work on CIRS-7was not conclusive and could not confirm a major implication of this pathway in Imid resistance. 

I would maybe propose a publication in one of the sister journals, such as International Journal of Molecular Sciences.

Author Response

We thank the reviewer for this suggestion; however, this paper is part of a special issue on multiple myeloma and should therefore be published in Cancers.

Reviewer 4 Report

The authors performed an exploratory investigation to identify circRNAs responsible for the IMiD-resistance of MM. Their screening experiments comparing circRNA expression profiles between IMiD-resistance and -sensitive MM selected several candidates including ciRS-7 that significantly down-regulated in IMiD-resistant MM.  This finding led the authors to hypothesize that down-regulation of ciRS-7 would play a causal role in increasing IMiD-resistance.  To test the hypothesis, the authors generated ciRS-7 KO MM cells, which failed to show altered IMiD-resistance.  

This reviewer acknowledges the willingness of the journal to publish "meaningful but negative" results. In order for this negative data to be more meaningful, the authors should test how the overexpression of ciRS-7 would affect the IMiD-resistance.

Author Response

We thank the reviewer for the comments and for agreeing with us that we have presented meaningful but negative results for ciRS-7. However, we do believe that our paper also include a number of very interesting findings that warrant publication in Cancers

This reviewer suggests that we also perform an overexpression of ciRS-7 and assess what effects this may have on IMiD sensitivity. However, this type of experiment is not straightforward to carry out, nor to interpret. We have previously established an overexpression system for ciRS-7, but the expression vector gave rise to both circular and linear products as well as concatemers (assessed by Northern blotting +/- RNase R). Of note, these are problems, which are largely overlooked in the circular RNA research field, despite giving rise to data that can be difficult to draw meaningful conclusions from. Moreover, the levels of the overexpression are much higher than endogenous ciRS-7 expression levels. For these reasons, it is difficult to decipher if potential phenotypic changes, as a result of such an overexpression system, should be contributed to the circular RNA, the by-products or the unnaturally high expression levels. In addition, it is well known that cells derived from B-cell malignancies are notoriously hard to transfect with larger constructs (we have previously tried several strategies without luck). Therefore, even if we would succeed in establishing an overexpression of ciRS-7 in the IMiD resistant cells, which we foresee may take 6-12 months as we currently do not have IMiD resistant cells running and are under lockdown, we believe that these data would not improve the paper because they would be largely inconclusive.

Round 2

Reviewer 1 Report

I value as positive that the authors have explained thoroughly the reasons why they defend the importance of their work in the field despite the negative results, and also that they have argued well the reasons why it is not feasible to carry out the suggested experiments.

Reviewer 4 Report

This reviewer is satisfied with the response by the authors and endorses the publication in Cancers.

This manuscript is a resubmission of an earlier submission. The following is a list of the peer review reports and author responses from that submission.

Round 1

Reviewer 1 Report

Jakobsen et al analyze genome-wide circRNA expression patterns in an in vitro model of acquired resistance to IMiDs (lenalidomide and pomalidomide) generated from the NCI-H929 cell line. They observe that specific circRNA patterns are associated with IMiD sensitive (parental NCI-H929 and LR and PR treated with EZH2i or EZH2i + DNMTi) or resistant (untreated LR and PR as well as LR and PR treated with panobinostat or DNMTi) cells. Although they find that ciRS-7 is epigenetically silenced in resistant cells, when they perform functional assays (knockdown of ciRS-7 in sensitive cells) they do not observe that sensitive cells become more resistant concluding that ciRS-7 does not seem to be directly involved in mediating resistance to IMiDs.

General comments:

The topic of the work is interesting, however, some weak points are found. Mainly, a great part of the work is just descriptive since the authors comment on the number of circRNAs deregulated in resistant cells vs sensitive cells and performed different types of statistical analysis and representations of the data to identify global differences and similarities between sensitive and resistant cells. Moreover, when they focus on ciRS-7 and carry out several experiments to unravel its potential implication in acquired resistance they conclude that it does not seem to be directly involved in mediating such resistance. Therefore, with the provided data it is difficult to predict a clear impact of all these results in the clinic.

Specific comments:

  • With respect to what it is mentioned in lines 176 to 181, I would suggest to carry out functional studies with circCDYL, circZKSCAN1 and circHIPK3 since, as commented by the authors, the expression of these circRNAs changes after treatment with epi-drugs (at least in LR cells).

  • Did the authors do some assays to evaluate if circHIPK3 (downregulated in LR cells) is epigenetically silenced (similarly to ciRS-7)? In addition, did they study whether silencing of circHIPK3 in sensitive cells induce to resistance?

  • Since authors observe that ciRS-7 is epigenetically silenced after prolonged cell culture without drug-exposure, my suggestion is to reanalyze data using NCI-H929 cells cultured during the same long period of time as LR and PR cells as a control cell line.

  • I would suggest, if possible, performing the same studies with other MM cell lines with acquired IMiD-resistance generated from sensitive cells (for example, MM.1S or OPM-2) in order to identify potential common circRNA patterns among different types of resistant cell lines.

Reviewer 2 Report

The paper presents interesting regults related to the important role of cirRNA in pathological status, like modulation of drug response in multiple myeloma cells.

the main limitaion of the study is related to  reducing number of samples used for cirRNA proffiling, to avoid this limitation additional validation of the most relevant altered cirRNAs can be performed.

Lack of characterization clear presentation of expression levels for cirRNAs in cell lines, particular for ciRS-7.

Data from cell lines and biological specimens should be presented separatelly, used venn diagrams for emphasis common and specific circRNA signature, particular those related to lenalidomide-resistant or pomalidomide-resistant.

Figure 6 demonstrates the utilization of the controls for transfection,  but it is not clear marked the scrambled control.

It is not clear about the mechanistic effect of inhibition of ciRS-7, this can be added in a new figure to summarize all the findings and implication of inhibition of this important circRNA.

Reviewer 3 Report

This is an elegant study on the contribution of circular RNA in mediating Imid resistance.

Comments

In figure 1, when presenting the heatmaps, the clustering is not clear to me and does not show differences between resistant vs sens cells. Can the authors show only parental cells and Imid resistant cells and not all the epi-treated cells?

Can the authors disclose the list and chromosomal location of circular RNAs identified in wt and IMID resistant cells? This is important for the scientific community working on non-coding RNA and circular RNA

The authors identified CIRS-7 as downl-regulated cRNA (through methylation) and these data are solid. However the implication and importance of this dwonregulation are less clear and should be further elaborated. No effects were after silencing. no putative mechanism could be identified or confirmed.

Minor remark: do the parental and Imid-resistant cells come from the same lab?

Can the authors disclose the list and chromosomal location of circular RNAs identified in wt and IMID resistant cells? This is important for the scientific community working on non-coding RNA and circular RNA